# An Ensemble of Retrieval-Based and Generation-Based Human-Computer Conversation Systems

## Abstract

Human-computer conversation systems have attracted much attention in Natural Language Processing. Conversation systems can be roughly divided into two categories: retrieval-based and generation-based systems. Retrieval systems search a user-issued utterance (namely a *query*) in a large conversational repository and return a reply that best matches the query. Generative approaches synthesize new replies. Both ways have certain advantages but suffer from their own disadvantages. We propose a novel ensemble of retrieval-based and generation-based conversation system. The retrieved candidates, in addition to the original query, are fed to a reply generator via a neural network, so that the model is aware of more information. The generated reply together with the retrieved ones then participates in a re-ranking process to find the final reply to output. Experimental results show that such an ensemble system outperforms each single module by a large margin.

## 1 Introduction

Automatic human-computer conversation systems have long served humans in domain-specific scenarios. A typical approach for such systems is built by human engineering, for example, using manually constructed ontologies (Wen et al. (2015)), natural language templates (Su et al. (2016)), and even predefined dialogue state tracking (Williams et al. (2013)).

Recently, researchers have paid increasing attention to open-domain, chatbot-style human-computer conversations such as XiaoIce[1] and Duer[2] due to their important commercial values. For open-domain conversations, rules and templates would probably fail since they hardly can handle the great diversity of conversation topics and flexible representations of natural language sentences. With the increasing popularity of on-line social media and community question-answering platforms, a huge number of human-human conversation utterances are available on the public Web (Yan et al. (2016a); Li et al. (2016b)). Previous studies begin to develop data-oriented approaches, which can be roughly categorized into two groups: retrieval systems and generative systems.

When a user issues an utterance (called a *query*), the retrieval-based conversation systems search a corresponding utterance (called a *reply*) that best matches the query in a pre-constructed conversational repository (Isbell et al. (2000); Ji et al. (2014)). Owing to the abundant web resources, the retrieval mechanism will always find a candidate reply given a query using semantic matching. The retrieved replies usually have various expressions with rich information. However, the retrieved replies are limited by the capacity of the pre-constructed repository. Even the best matched reply from the conversational repository is not guaranteed to be a good response since most cases are not tailored for the issued query.

To make a reply tailored appropriately for the query, a better way is to generate a new one accordingly. With the prosperity of neural networks powered by deep learning, generation-based conversation systems are developing fast. Generation-based conversation systems can synthesize a new sentence as the reply, and thus bring the results of good flexibility and quality. A typical generation-based conversation model is `seq2seq` (Sordoni et al. (2015); Shang et al. (2015); Serban et al. (2016a)), in which two recurrent neural networks (RNNs) are used as the encoder and the decoder. The encoder is to capture the semantics of the query with one or a few distributed and real-valued vectors (also known as *embeddings*); the decoder aims at decoding the query embeddings to a reply. Long short term memory (LSTM) (Hochreiter & Schmidhuber (1997)) or gated recurrent units

---

[1] http://www.msxiaoice.com/
[2] http://duer.baidu.com/

| Category | Pros | Cons |
|---|---|---|
| Retrieval | literal human utterances; various expressions with great diversity; | not tailored to queries; bottleneck is the size of repository |
| Generation | tailored for queries; highly coherent | insufficient information; universal sentences |

Table 1: Characteristics of retrieved and generated replies in two different conversational systems.

(GRUs) (Cho et al. (2014)) could further enhance the RNNs to model longer sentences. The advantage of generation-based conversation systems is that they can produce flexible and tailored replies. A well known problem for the generation conversation systems based on "Seq2Seq" is that they are prone to choose universal and common generations. These generated replies such as "I don't know" and "Me too" suit many queries (Serban et al. (2016a)), but they contain insufficient semantics and information. Such insufficiency leads to non-informative conversations in real applications.

Previously, the retrieval-based and generation-based systems with their own characteristics, as listed in Table 1, have been developed separately. We are seeking to absorb their merits. Hence, we propose an ensemble of retrieval-based and generation-based conversation systems. Specifically, given a query, we first apply the retrieval module to search for $k$ candidate replies. We then propose a "multi sequence to sequence" (multi-seq2seq) model to integrate each retrieved reply into the Seq2Seq generation process so as to enrich the meaning of generated replies to respond the query. We generate a reply via the multi-seq2seq generator based on the query and $k$ retrieved replies. Afterwards, we construct a re-ranker to re-evaluate the retrieved replies and the newly generated reply so that more meaningful replies with abundant information would stand out. The highest ranked candidate (either retrieved or generated) is returned to the user as the final reply. To the best of our knowledge, we are the first to build a bridge over retrieval-based and generation-based modules to work out a solution for an ensemble of conversation system.

Experimental results show that our ensemble system consistently outperforms each single component in terms of subjective and objective metrics, and both retrieval-based and generation-based methods contribute to the overall approach. This also confirms the rationale for building model ensembles for conversation systems.

## 2 RELATED WORK

In early years, researchers mainly focus on domain-specific conversation systems, e.g., train routing (Aho & Ullman (1972)) and human tutoring (Graesser et al. (2005)). Typically, a pre-constructed ontology defines a finite set of slots and values, for example, cuisine, location, and price range in a food service conversation system; during human-computer interaction, a state tracker fills plausible values to each slot from the user input, and recommend the restaurant that best meets the user's requirement (Williams (2014); Mrkšić et al. (2015); Wen et al. (2016)).

In the open domain, however, such slot-filling approaches would probably fail because of the diversity of topics and natural language utterances. Isbell et al. (2000) apply information retrieval techniques to search for related queries and replies. Ji et al. (2014) and Yan et al. (2016a) use both shallow hand-crafted features and deep neural networks for matching. Li et al. (2016b) propose a random walk-style algorithm to rank candidate replies. In addition, their model can incorporate additional content (related entities in the conversation context) by searching a knowledge base when a stalemate occurs during human-computer conversations.

Generative conversation systems have recently attracted increasing attention in the NLP community. Ritter et al. (2011) formulate query-reply transformation as a phrase-based machine translation. Zoph & Knight (2016) use two RNNs in encoder and one RNN in decoder to translate a sentence into two different languages into another language. Since the last year, the renewed prosperity of neural networks witnesses an emerging trend in using RNN for conversation systems (Sutskever et al. (2014); Vinyals & Le (2015); Sordoni et al. (2015); Shang et al. (2015); Serban et al. (2016a)). The prevalent structure is the seq2seq model (Sutskever et al. (2014)) which comprises of one encoder and one decoder. However, a known issue with RNN is that it prefers to generate short and meaningless utterances. Following the RNN-based approach, Li et al. (2016a) propose a mutual information objective in contrast to the conventional maximum likelihood criterion. Mou et al. (2016) and Xing et al. (2016) introduce additional content (i.e., either the most mutually informative word or the topic information) to the reply generator. Serban et al. (2016b) applies a variational encoder to capture query information as a distribution, from which a random vector is sampled for reply

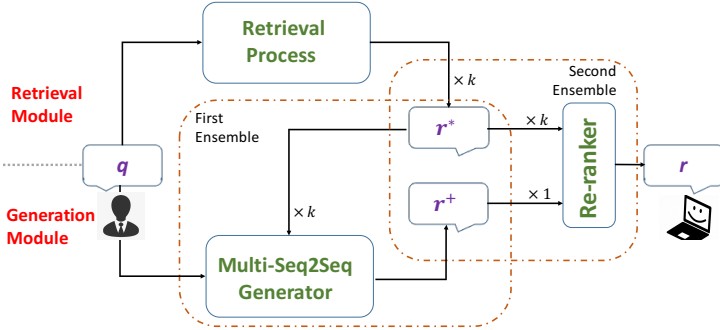

Figure 1: The overall architecture of our model ensemble. We combine retrieval-based and generation-based conversation systems with two mechanisms. The first ensemble is to enhance the generator with the retrieved candidates. The second is the re-ranking of both candidates.

generation. He et al. (2017) uses knowledge base for answer generation in question answering task and Libovicky & Helcl (2017) investigates different attention strategies in multi-source generation.

To the best of our knowledge, the two main streams namely retrieval-based and generation-based systems have developed independently, and we are the first to combine these two together. In the following section, we depict the whole picture of our ensemble framework, and describe how to integrate those two modules in detail.

# 3 MODEL ENSEMBLE

## 3.1 OVERVIEW

Figure 1 depicts the overview of our proposed conversation system that ensembles the retrieval-based and generation-based approaches. It consists of the following components. We briefly describe each component, then present the details in the following sub-sections.

• **Retrieval Module.** We have a pre-constructed repository consisting millions of query-reply pairs $\langle q^*, r^* \rangle$, collected from human conversations. When a user sends a query utterance $q$, our approach utilizes a state-of-the-practice information retrieval system to search for $k$ best matched queries ($q^*$), and return their associated replies $r^*$ as $k$ candidates.

• **Generation Module.** We propose the `multi-seq2seq` model, which takes the original query $q$ and $k$ retrieved candidate replies $r_1^*, r_2^*, \ldots, r_k^*$ as input, and generates a new reply $r^+$. Thus the generation process could not only consider about the given query, but also take the advantage of the useful information from the retrieved replies. We call it the **first ensemble** in our framework.

• **Re-ranker.** Finally, we develop a re-ranker to select the best reply $r$ from the $k+1$ candidates obtained from retrieval-based and generation-based modules. Through the ensemble of retrieval-based and generation-based conversation, the enlarged candidate set enhances the quality of the final result. We call this procedure the **second ensemble** in our framework.

## 3.2 RETRIEVAL-BASED CONVERSATION SYSTEM

The information retrieval-based conversation is based on the assumption that the appropriate reply to the user's query is contained by the pro-constructed conversation datasets. We collect huge amounts of conversational corpora from on-line chatting platforms, whose details will be described in the section of evaluation. Each utterance and its corresponding reply form a pair, denoted as $\langle q^*, r^* \rangle$.

Based on the pre-constructed dataset, the retrieval process can be performed using an the state-of-the-practice information retrieval system. We use a `Lucene` [3] powered system for the retrieval implementation. We construct the inverted indexes for all the conversational pairs at the off-line stages. When a query is issued, the keyword search with the *tf.idf* weighting schema will be performed to retrieve several $q^*$ that match the user's query $q$. Given the retrieved $q^*$, the associated $r^*$ will be returned as the output, result in an indirect matching between the user's query $q$ and the retrieved reply $r^*$. The retrieval systems would provide more than one replies and score them according to the semantic matching degree, which is a traditional technic in information retrieval. As the top ranked one may not perfectly match the query, we keep top-$k$ replies for further process.

The information retrieval is a relatively mature technique, so the retrieval framework can be alternated by any systems built keep to the following principles.

---

[3]http://lucene.apache.org

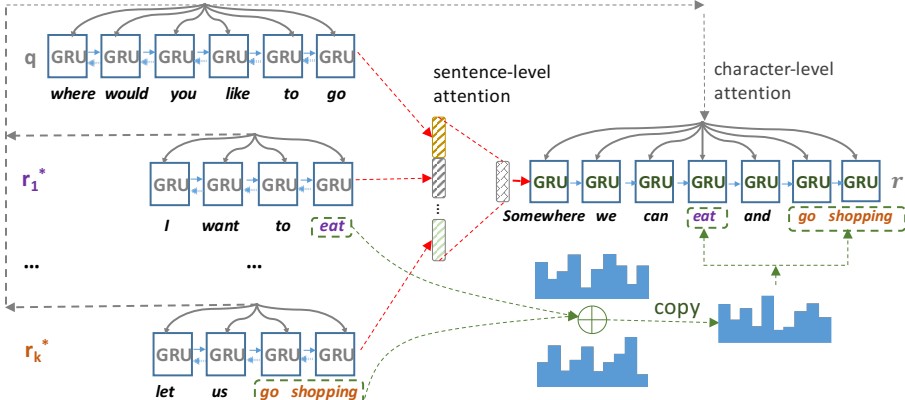

Figure 2: The `multi-seq2seq` model, which takes a query $q$ and $k$ retrieved candidate replies $r^*$ as the input and generate a new reply $r^+$ as the output.

### 3.3 GENERATION-BASED CONVERSATION SYSTEM

The technique of neural networks has become a popular approach to build end-to-end trainable conversation systems (Mou et al. (2016)). A generation-based conversation system is able to synthesize new utterances, which is complementary to retrieval-based methods.

The `seq2seq` model (Sutskever et al. (2014)), considering the Recurrent Neural Network (RNNs) as the encoder and decoder to transfer source sentence to target sentence, has long been used for generation tasks. The objective function for the `seq2seq` model in our scenario is the log-likelihood of the generated reply $r^+$ given the query $q$:

$$\hat{r^+} = \underset{r^+}{\operatorname{argmax}} \left\{ \log p(r^+|q) \right\} \tag{1}$$

Since the reply is generated on the conditional probabilities given the query, the universal replies which have relatively higher probabilities achieve higher rankings. However, these universal sentences contain less information, which impair the performance of generative systems. Mou et al. (2016) also observe that in open-domain conversation systems, if the query does not carry sufficient information, `seq2seq` tends to generate short and meaningless sentences.

Different from the pipeline in `seq2seq` model, we propose the `multi-seq2seq` model (Figure 2), which synthesizes a tailored reply $r^+$ by using the information both from the query $q$ and the retrieved $r_1^*, r_2^*, \ldots, r_k^*$. `multi-seq2seq` employs $k+1$ encoders, one for query and other $k$ for retrieved $r^*$. The decoder receives the outputs of all encoders, and remains the same with traditional `seq2seq` for sentence generation. `multi-seq2seq` model improves the quality of the generated reply in two ways. First, the newly generated reply conditions not only on the given query but also on the retrieved reply. So the probability of universal replies would decrease since we add an additional condition. The objective function can be written as:

$$\hat{r^+} = \underset{r^+}{\operatorname{argmax}} \left\{ \log p(r^+|q, r_1^*, r_2^*, \ldots, r_k^*) \right\} \tag{2}$$

Thus the $r^+$ would achieve higher score only if it has a high concurrency with both $q$ and $r_1^*, r_2^*, \ldots, r_k^*$. Second, the retrieved replies $r_1^*, r_2^*, \ldots, r_k^*$ are the human-produced utterances and probably contain more information, which could be used as the additional information for the generated reply $r^+$. Hence, the generated reply can be fluent and tailored to the query, and be more meaningful due to the information from the retrieved candidates. To take advantage of retrieved replies, we propose to integrate attention and copy mechanisms into decoding process. Attention helps the decoder to decide which parts in each retrieved reply are useful for current generation step. Copy mechanism directly extracts proper words from encoders, namely both query and retrieved replies, and utilizes them as the output words during the decoding process.

● **Two-level Attention.** `multi-seq2seq` conducts sentence- and character- level attention to make better use of the query and retrieved replies. As multiple replies are of uneven quality, we use sentence-level attention to assign different importance to each retrieved replies. Similarly, multiple words are of uneven quality in a sentence, we use character-level attention to measure different importance to each words in retrieved replies. Specifically, for the sentence-level, we use $k+1$ vectors obtained from the encoders to capture the information of $q$ and the $k$ $r^*$, denoted as $\boldsymbol{q}$ and $\boldsymbol{r}_1^* \ldots, \boldsymbol{r}_k^*$, which are concatenated as $[\boldsymbol{q}; \boldsymbol{r}^*; \ldots; \boldsymbol{r}_k^*]$. This vector is linearly transformed before fed to the decoder as the initial state. For the character-level, we extend the traditional attention (Bahdanau et al.

(2015)) to multi-source attention to introduce retrieved replies, given by

$$c_i = \sum_{j=1}^{l} \alpha_{i,j} \boldsymbol{h}_j + \sum_{m=1}^{k} \sum_{j=1}^{l_m} \alpha_{i,m,j} \boldsymbol{h}_{m,j} \tag{3}$$

$$\alpha_{i,m,j} = \frac{\exp e_{i,m,j}}{\sum_{j=1}^{l_m} e_{i,m,j}}, e_{i,m,j} = \tanh(\boldsymbol{s}_{i-1} M_a \boldsymbol{h}_{m,j}) \tag{4}$$

where $c_i$ is the context vector at each time step in decoding, which integrates query and all possible words in $k$ retrieved replies. $l$ is the length of query, $\boldsymbol{h}_j$ is the hidden state of query, $l_m$ is the length of $r_m^*$, $\boldsymbol{h}_{m,j}$ is the hidden state of $r_m^*$. $\boldsymbol{s}_i$ is the hidden state of decoder at time step $i$, $\alpha_{i,m,j}$ is the normalized attention weights for each word. $e_{i,m,j}$ is calculated by a bilinear matching function and $M_a$ is the parameter matrix.

• **Copy Mechanism.** `multi-seq2seq` also uses copy mechanism to explicitly extract words from the retrieved replies. For each word $y_t$ in vocabulary $V$, the probability $p(y_t|\boldsymbol{s}_t)$ in decoding process is comprised of $k+1$ parts. The first part $p_{ori}$ follows the original probability calculated by GRU/LSTM cells, and the following parts $p_{r_m^*}$ reflect the matching degree between the current state vector $\boldsymbol{s}_t$ and the corresponding states of $y_t$ in encoders, given by,

$$p(y_t|\boldsymbol{s}_t) = p_{ori}(y_t|\boldsymbol{s}_t) + \sum_{m=1}^{k} p_{r_m^*}(y_t|\boldsymbol{h}_{y_t,m}) \tag{5}$$

$$p_{r_m^*}(y_t|\boldsymbol{h}_{y_t,m}) = \delta(\boldsymbol{s}_t M_c \boldsymbol{h}_{y_t,m}) \tag{6}$$

where $\boldsymbol{h}_{y_t,m}$ is the hidden states of retrieved reply $r_m^*$ who responds $y_t$ in decoder, $\delta(\cdot)$ is the sigmoid function, $M_c$ is the parameter for matching $\boldsymbol{s}_t$ and $\boldsymbol{h}_{y_t,m}$. If $y_t$ has not appeared in a retrieved replies $r_m^*$, the corresponding probabilities $p_{r_m^*}$ would be zero.

Both attention and copy mechanism aim to enrich the generated reply $r^+$ via useful and informative words extracted from retrieved replies $r_1^*, r_2^*, \ldots, r_k^*$. Figure 2 displays the design of `multi-seq2seq` model. We can see that the generated reply has the corresponding relation with the query, and absorbs the keywords from the retrieved replies.

## 3.4 RE-RANKER

Now that we have $k$ retrieved candidate replies $r^*$ as well as a generated one $r^+$. As all the retrieved candidates are obtained via indirect matching, these replies need a further direct matching with the user-issued query. On the other hand, the generated reply set may contain the influent and meaningless utterances. Hence, we propose the second ensemble to derive the final ranking list by feeding all the candidates into a re-ranker.

We deploy a Gradient Boosting Decision Tree (GBDT) (Ye et al. (2009)) classifier since it is believed to have ability to handle the replies with various traits. The GBDT classifier utilizes several high-level features, as listed in the following. The first four are pairwise feature, and the last two are features based on the properties of replies.

• *Term similarity.* The word overlap ratio captures the literal similarity between the query and reply. For both query and reply, we transform them into binary word vectors, in which each element indicates if a word appears in the corresponding sentence. Then we apply the cosine function to calculate the term overlap similarity of the query and the reply.

• *Entity similarity.* Named entities in utterances are a special form of terms. We distinguish *persons*, *locations* and *organizations* from plain texts with the help of named entity recognition techniques. Then we maintain the vectors of recognized entities for both query and its reply and calculate the similarity (measured by cosine similarity) between two entity-based vector representations.

• *Topic similarity.* "Topics" has long been regarded as the abstractive semantic representation (Hofmann (2001)). We apply Latent Dirichlet Allocation (Blei et al. (2003)) to discover the latent topics of the query and reply. The inferred topic representation is the probabilities for the piece of text belonging to each latent topic. By setting the topic number as $1000$, which works efficiently in practice, we use the cosine similarity to calculate the topical score between vectors of latent topics.

• *Statistical Machine Translation.* By treating queries and replies as different languages in the paradigm of machine translation, we train a translation model to "translate" the query into a reply based on the training corpora to get the translating word pairs (one word from a query and one word from its corresponding reply) with scores indicating their translating possibilities. To get

| Dataset | # of samples |
|---|---|
| Retrieval (Repository) | 7,053,820 |
| Re-ranker (Train) | 50,000 |
| Generator (Train) | 1,500,000 |
| Validation | 100,000 |
| Testing | 6,741 |

Table 2: Statistics of our datasets.

the translation score for the query and reply, we sum over the translating scores of the word pairs extracted from these two sentences, and conduct normalization on the final score.

• *Length.* Since too short replies are not preferred in practical conversation systems, we take the length of replies as a point-wise feature. We conduct a normalization to map the value to [0,1].

• *Fluency.* Fluency is to examine whether two neighboring terms have large co-occurrence likelihood. We calculate the co-occurrence probability for the bi-grams of the candidate replies and then take the average value as the fluency.

The confidence scores produced by the GBDT classifier are used to re-rank all the replies. The re-ranking mechanism can eliminate both meaningless short replies that are eventually generated by `multi-seq2seq` and less appropriate replies selected by the retrieval system. The *re-ranker* further ensures an optimized effect of model ensemble.

### 3.5 MODEL TRAINING

Since our framework consists of learnable but independent components (i.e., `multi-seq2seq` and Re-ranker), the model training is constructed for each component separately.

In `multi-seq2seq`, we use human-human utterance pairs $\langle q, r \rangle$ as data samples. $k$ retrieved candidates $r^*$ are also provided as the input when we train the neural network. Standard cross-entropy loss of all words in the reply is applied as the training objective, given by,

$$J = -\sum_{i=1}^{T} \sum_{j=1}^{V} t_j^{(i)} \log y_j^{(i)} \tag{7}$$

where $J$ is the objective of trainning, $T$ is the length of $r$ and $\boldsymbol{t}^{(i)}$ is the one-hot vector of the next target word in the reply, serving as the ground-truth, $y_j$ is the probability of a word obtained from the softmax function, and $V$ is the vocabulary size.

In the re-ranker part, the training samples are either $\langle q, r \rangle$ pairs or generated by negative sampling.

## 4 EVALUATION

We evaluate our ensemble model on our established conversation system in Chinese.

### 4.1 EXPERIMENTAL SETUP

Both retrieval-based and generation-based components require a large database of query-reply pairs, whose statistics is exhibited in Table 2. To construct a database for information retrieval, we collected human-human utterances from massive online forums, microblogs, and question-answering communities, including Sina Weibo[4] and Baidu Tieba.[5] In total, the database contains 7 million query-reply pairs for retrieval. For each query, corresponding to a question, we retrieve $k$ replies ($k = 2$) for generation part and re-ranker.

For the generation part, we use the dataset comprising 1,606,741 query-reply pairs originating from Baidu Tieba. Please note that $q$ and $r^*$ are the input of `multi-seq2seq`, whose is supposed to should approximate the ground-truth. We randomly selected 1.5 million pairs for training and 100K pairs for validation. The left 6,741 pairs are used for testing both in generation part and the whole system. Notice that this corpus is different from the corpus used in the retrieval part so that the ground-truth of the test data are excluded in the retrieval module. The training-validation-testing split remains the same for all competing models.

---

[4] http://weibo.com
[5] http://tieba.baidu.com

| Method | Human Score | BLEU-1 | BLEU-2 | BLEU-3 | BLEU-4 |
|---|---|---|---|---|---|
| Retrieval-1 | 1.013 | **24.06** | 10.04 | 5.232 | 2.784 |
| Retrieval-2 | 0.528 | 4.532 | 0.6549 | 0.4775 | 0.4708 |
| seq2seq | 0.880 | 6.349 | 0.6647 | 0.1105 | 0.0393 |
| Ensemble(retrieval-1, retrieval-2, seq2seq) | 1.145 | 14.15 | 8.40 | 7.798 | 7.619 |
| multi-seq2seq⁻ | 0.9180 | 9.290 | 2.489 | 1.144 | 0.5660 |
| multi-seq2seq | 0.9920 | 9.609 | 1.674 | 0.5100 | 0.1911 |
| Ensemble(retrieval-1, retrieval-2, multi-seq2seq) | **1.362** | 16.991 | **11.133** | **10.37** | **9.993** |

Table 3: Results of our ensemble and competing methods in terms of average human scores and BLEUs. Inter-annotator agreement for human annotation: Fleiss' $\kappa = 0.2932$ (Fleiss (1971)), std $= 0.3926$, indicating moderate agreement. While the agreement is comparable to previous results, e.g., 0.2–0.4 reported in Shang et al. (2015)

To train our neural models, we implement code based on dl4mt-tutorial[6], and follow Shang et al. (2015) for hyper-parameter settings as it generally works well in our model. We did not tune the hyperparameters, but are willing to explore their roles in conversation generation in future. All the embeddings are set to 620-dimension and the hidden states are set to 1000-dimension. We apply AdaDelta with a mini-batch (Zeiler (2012)) size of 80. Chinese word segmentation is performed on all utterances. We keep the same set of 100k words for all encoders and 30K for the decoder due to efficiency concerns. The validation set is only used for early stop based on the perplexity measure.

### 4.2 COMPETING METHODS

We compare our model ensemble with each individual component and provide a thorough ablation test. Listed below are the competing methods in our experiments. For each method, we keep one best reply as the final result to be assessed. All competing methods are trained in the same way as our full model, when applicable, so that the comparison is fair.

• *Retrieval-1, Retrieval-2*. The top and second ranked replies for the user-issued query from a state-of-the-practice conversation system (Yan et al. (2016b)), which is a component of our model ensemble; it is also a strong baseline (proved in our experiments).

• seq2seq. An encoder-decoder framework (Sutskever et al. (2014)), first introduced as neural responding machine by Shang et al. (2015).

• multi-seq2seq⁻. Generation component, which only applies two-level attention strategies.

• multi-seq2seq. Generation component, which applies two-level attention and copy strategy.

• *Ensemble(Retrieval-1,Retrieval-2, seq2seq)*. Ensemble with retrieval and seq2seq.

• *Ensemble(Retrieval-1, Retrieval-2, multi-seq2seq)*. Ensemble with retrieval and multi-seq2seq. This is the full proposed model ensemble.

### 4.3 OVERALL PERFORMANCE

We evaluate our approach in terms of both subjective and objective metrics.

• Subjective metric. Human evaluation, albeit time- and labor-consuming, conforms to the ultimate goal of open-domain conversation systems. We ask three educated volunteers to annotate the results (Shang et al. (2015); Li et al. (2016b); Mou et al. (2016)). Annotators are asked to label either "0" (bad), "1" (borderline), or "2" (good) to a query-reply pair. The subjective evaluation is performed in a strictly random and blind fashion to rule out human bias.

• Objective metric. We adopt BLEU 1-4 for the purpose of automatic evaluation. While Liu et al. (2016) further strongly argue that no existing automatic metric is appropriate for open-domain dialogs, they show a slight positive correlation between BLEU-2 and human evaluation in non-technical Twitter domain, which is similar to our scenario. We nonetheless include BLEU scores as the expedient objective evaluation, serving as a supporting evidence. BLEUs are also used in Li et al. (2016a) for model comparison and in Mou et al. (2016) for model selection.

Notice that, the automatic metrics were computed on the entire test set, whereas the subjective evaluation was based on 100 randomly chosen test samples due to the limitation of human resources.

We present our main results in Table 3. Table 4 presents two examples of our ensemble and its "base" models. As showed, the retrieval system, which our model ensemble is based on, achieves better performance than RNN-based sequence generation. This also verifies that the retrieval-based conversation system in our experiment is a strong baseline to compare with.

---

[6]https://github.com/nyu-dl/dl4mt-tutorial

| | Utterance (Translated) | Selected by re-ranker |
|---|---|---|
| Query | This mobile phone's photo effect is pretty good. | |
| Retrieved-1 | I really have a crush on it. | |
| Retrieved-2 | Go for it. | |
| multi-seq2seq | Rushing for it rather than having a crush on it. | √ |
| seq2seq | Ha-ha. | |
| Query | Can I see the house tomorrow afternoon? | |
| Retrieved-1 | You can call me! | |
| Retrieved-2 | You can see the house on weekends. | |
| multi-seq2seq | You can see the house on weekends, please call me in advance. | √ |
| seq2seq | OK. | |

Table 4: Examples of retrieved and generated ones. "√" indicates the reply selected by the re-ranker.

Combining the retrieval system, generative system `multi-seq2seq` and the re-ranker, our model leads to the bset performance in terms of both human evaluation and BLEU scores. Concretely, our model ensemble outperforms the state-of-the-practice retrieval system by $+34.45\%$ averaged human scores, which we believe is a large margin.

### 4.4 ANALYSIS AND DISCUSSION

Having verified that our model achieves the best performance, we are further curious how each gadget contributes to our final system. Specifically, we focus on the following research questions.

**RQ1**: What is the performance of `multi-seq2seq` (the **First Ensemble** in Figure 1) in comparison with traditional `seq2seq`?

From BLEU scores in Table 3, we see both `multi-seq2seq`⁻ and `multi-seq2seq` significantly outperform conventional `seq2seq`, and `multi-seq2seq` is slightly better than `multi-seq2seq`⁻. These results imply the effectiveness of both two-level attention and copy mechanism. We can also see `multi-seq2seq` outperforms the second retrieval results in BLEUs. In the retrieval and `seq2seq` ensemble, 72.84% retrieved and 27.16% generated ones are selected. In retrieval and `multi-seq2seq` ensemble, the percentage becomes 60.72% vs. 39.28%. The trend indicates that `multi-seq2seq` is better than `seq2seq` from the re-ranker's point of view.

**RQ2**: How do the retrieval- and generation-based systems contribute to re-ranking (the **Second Ensemble** in Figure 1)?

As the retrieval and generation module account for 60.72% and 39.28% in the final results of retrieval and `multi-seq2seq` ensemble, they almost contribute equally to the whole framework. More importantly, we notice that retrieval-1 takes the largest proportion in two ensemble systems, and it may explain why most on-line chatting platforms choose retrieval methods to build their systems. `multi-seq2seq` decreases the proportion of retrieved one in the second ensemble systems, which indicates `multi-seq2seq` achieves better results than retrieval-1 in some cases.

**RQ3**: Since the two ensembles are demonstrated to be useful already, can we obtain further gain by combining them together?

We would also like to verify if the combination of `multi-seq2seq` and re-ranking mechanisms yields further gain in our ensemble. To test that, we compare the full model *Ensemble(Retrieval, multi-seq2seq)* with an ensemble that uses traditional `seq2seq`, i.e., *Ensemble(Retrieval, seq2seq)*. As indicated in Table 3, even with the re-ranking mechanism, the ensemble with underlying `multi-seq2seq` still outperforms the one with `seq2seq`. Likewise, *Ensemble(Retrieval, multi-seq2seq)* outperforms both *Retrieval* and *multi-seq2seq* in terms of most metrics.

Through the above ablation tests, we conclude that both components (first and second ensemble) play a role in our ensemble when we combine the retrieval- and generation-based systems.

### 5 CONCLUSION

In this paper, we propose a novel ensemble of retrieval-based and generation-based open-domain conversation systems. The retrieval part searches the $k$ best-matched candidate replies, which are, along with the original query, fed to an RNN-based `multi-seq2seq` reply generator. Then the generated replies and retrieved ones are re-evaluated by a re-ranker to find the final result. Although traditional generation-based and retrieval-based conversation systems are isolated, we have designed a novel mechanism to connect both modules. The proposed ensemble model clearly outperforms state-of-the-art conversion systems in the constructed large-scale conversation dataset.

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
