# OpenReview forum: "An Ensemble of Retrieval-Based and Generation-Based Human-Computer Conversation Systems."
_ICLR.cc/2018/Conference — Reject_

### Official Review · AnonReviewer1 · 2017-11-27
**Interesting combination of retrieval and generation based modules**

**Rating:** 5
**Confidence:** 3

**Review:**

Summary:

The paper proposes a new dialog model combining both retrieval-based and generation-based modules. Answers are produced in three phases: a retrieval-based model extracts candidate answers; a generator model, conditioned on retrieved answers, produces an additional candidate; a reranker outputs the best among all candidates.

The approach is interesting: the proposed ensemble can improve on both the retrieval module and the generation module, since it does not restrict modeling power (e.g. the generator is not forced to be consistent with the candidates). I am not aware of similar approaches for this task. One work that comes to mind regarding the blend of retrieval and generation is Memory Networks (e.g. https://arxiv.org/pdf/1606.03126.pdf and references): given a query, a set of relevant memories is extracted from a KB using an inverted index and the memories are fed into the generator. However, the extracted items in the current work are candidate answers which are used both to feed the generator and to participate in reranking.

The experimental section focuses on the task of building conversational systems. The performance measures used are 1) a human evaluation score with three volunteers and 2) BLUE scores. While these methods are not very satisfying, effective evaluation of such systems is a known difficulty.

The results show that the ensemble outperforms the individual modules, indicating that: the multi-seq2seq models have learned to use the new inputs as needed and that the ranker is correlated with the evaluation metrics.

However, the results themselves do not look impressive to me: the subjective evaluation is close to the "borderline" score; in the examples provided, one is good, the other is borderline/bad, and the baseline always provides something very short. Does the LSTM work particularly poor on this dataset? Given that this is a novel dataset, I don't know what the state-of-the-art should be. Could you provide more insight? Have you considered adding a benchmark dataset (e.g. a QA dataset)?

Specific questions:

1. The paper motivates conditioning on the candidates in two ways. First, that the candidates bring additional information which the decoder can use (e.g. read from the candidates locations, actions, etc.). Second, that the probability of universal replies must decrease due to the additional condition. I think the second argument depends on how the conditioning is performed: if the candidates are simply appended to the input, the model can learn to ignore them.
2. The copy mechanism is a nice touch, encouraging the decoder to use the provided queries. Why not copy from the query too, e.g. with some answers reusing part of the query <"Where are you going?", "I'm going to the park">?
3. How often does the model select the generated answer vs. the extracted answers? In both examples provided the selected answer is the one merging the candidate answers.

Minor issues:
- Section 3.2: using and the state
- Section 3.2: more than one replies
- last sentence on page 3: what are the "following principles"?

---

### Official Review · AnonReviewer3 · 2017-11-28
**Limited evaluation**

**Rating:** 5
**Confidence:** 3

**Review:**

The authors present a generation-based neural dialog response model that takes a list of retrieved responses from a search engine as input. This novel multi-seq2seq approach, which includes attention and a pointer network, increases reply diversity compared to purely retrieval or generation-based models. The authors also apply a reranking-based approach to ensembling based on a gradient-boosted decision tree classifier.
But their multi-seq2seq model is not particularly well-justified with evaluations and examples (as compared with the reranking/ensemble, which is essentially a standard approach) and it's unclear whether it helps more than other recent approaches to response diversity.

Some additional points:
1. For several of the GBDT features, the approach chosen is unusual or perhaps outdated. In particular, the choice of a word-level MT-based metric rather than an utterance-level one, and the choice of a bigram-based fluency metric rather than one based on a more complete language model are puzzling and should be justified.
2. The authors report primarily comparisons to ablations of their own model, not to other recent work in dialog systems.
3. The human evaluation performance of a simple reranking ensemble between the authors' generation-based model and their retrieval-based model is significantly higher than multi-seq2seq, suggesting that multi-seq2seq may not be an especially powerful way to combine information from the two models.
4. The authors present only very limited examples (in Table 4) and one out of the two multi-seq2seq examples in that table is relatively nonsensical. When the original examples are non-English, papers should also include the original in addition to a translation.

---

### Official Review · AnonReviewer2 · 2017-11-29
**This paper presents an improved conversation system based on new heuristic ensemble combinations of existing methods, namely generative and retrieval-based systems. The problem of building and improving query response systems or chat bots is very relevant to the ICLR community. Overall it is a solid application paper.**

**Rating:** 6
**Confidence:** 3

**Review:**

The approach involves multiple steps.
On a high level the query is first used to retrieve k best matching response candidates. Then a concatenation of the query and the candidates are fed into a generative model to generate an additional artificial candidate.
In a final step, the k+1 candidates are re-ranked to report the final response.
Each of these steps involves careful engineering and for each there are some minor novel components.
Yet, not all of the steps are presented in complete technical detail.
Also, training corpora and human labeling of the test data do not seem to be publicly available.
Consequently, it would be hard to exactly reproduce the results of the paper.
Experimental validation also is relatively thin.
While the paper report both BLEU metrics and Fleiss kappa from a small-scale human test, the results are based on a single split of a single corpus into training, validation and test data.
While the results for the ensemble are reported to be higher than for the various components for almost all metrics, measures of spread/variance would allow the reader to better judge the degree and significance of improvement.

Minor:
The paper should be read by a native speaker, as it involves a number of minor grammar issues and typos.

---

### Decision · Program_Chairs · 2018-01-29
**ICLR 2018 Conference Acceptance Decision**

**Decision:**

Reject

**Comment:**

This paper presents an ensemble method for conversation systems, where a retrieval-based system is ensembled with a generation-based system.  The combination is done via a reranker.  Evaluation is done on one dataset containing query reply pairs with both BLEU and human evalutations.  The experimental results are good using the ensemble model.  Although this presents some novel ideas and may be useful for chatbots (not for goal oriented systems), the committee feels that the approach and the presented material does not have enough substance for publication at ICLR:  it will be interesting to evaluate this system in a goal oriented setting; many prior papers have built generation based conversation systems (1-step) -- this paper does not present any comparison with those papers.  Addressing these issues may strengthen the paper for a future venue.

---

> ### Public Comment · ~Sam_Brooks1 · 2019-06-08
> **Conversation systems**
>
> Conversation systems can be roughly divided into two categories: retrieval-based and generation-based systems. Retrieval systems search a user-issued utterance in a large conversational repository and return a reply that best matches the query.
> http://happy-wheelsgames.com/